# The Pattern of Injuries in the Emergency Room during the COVID-19 Pandemic

**DOI:** 10.3390/healthcare11101483

**Published:** 2023-05-19

**Authors:** Yunhyung Choi, Duk Hee Lee

**Affiliations:** Department of Emergency Medicine, College of Medicine, Ewha Womans University, Seoul 07985, Republic of Korea

**Keywords:** COVID-19, pandemic, trauma, injury, emergency, pattern, trend

## Abstract

Background: The coronavirus disease 2019 (COVID-19) pandemic has obviously caused a remarkable change in patients’ emergency department (ED) visits; however, data from multicenter studies are lacking. We aimed to present a comprehensive analysis of injury-related ED visits in Republic of Korea before and during the COVID-19 pandemic. Materials and Methods: Data from 23 tertiary hospitals based on Emergency Department-based Injury In-depth Surveillance were used for this retrospective cross-sectional study. A total of 541,515 ED visits (age ≥ 20 years) between 1 January 2018 and 31 December 2020 were included, and the trend of injuries related to motor vehicular accidents, falls, self-harm and suicide, assault, and poisoning were compared between the pre-COVID-19 time period and during the COVID-19 pandemic. Results: In the first year of the COVID-19 period, a decline in the number of ED visits was observed (41,275, 21%) compared to the previous year. Injuries caused by motor vehicles (36,332 in 2019 vs. 27,144 in 2020), falls and slips (61,286 in 2019 vs. 49,156 in 2020), assaults (10,528 in 2019 vs. 8067 in 2020), and poisonings (7859 in 2019 vs. 7167 in 2020) decreased, whereas self-harm and suicide (8917 in 2019 vs. 8911 in 2020) remained unchanged. The hospitalization (16.6% in 2019 vs. 18.8% in 2020) and ED mortality rate (0.6% in 2019 vs. 0.8% in 2020) also increased. Conclusion: The COVID-19 pandemic led to a decline in the overall number of trauma patients seeking medical care; however, the proportion of patients requiring hospitalization or intensive care unit admission increased, indicating more severe injuries among those who did seek care. Suicide attempt rates remained unchanged, highlighting the need for targeted care and support for vulnerable patients. During the pandemic, EDs had to continue to provide care to patients with medical emergencies unrelated to COVID-19, which requires a delicate and adaptable approach to ED operations. To manage the increased stress and workload caused by the pandemic, increased resources and support for healthcare workers were needed.

## 1. Introduction

On 30 January 2020, the World Health Organization (WHO) declared the spread of severe acute respiratory syndrome coronavirus 2 causing coronavirus disease (COVID-19) as a Public Health Emergency of International Concern [1], which had a significant impact on patients’ visits to EDs, bringing significant changes to daily life. The world did not have prior experience in handling unique circumstances such as the COVID-19 pandemic, which led to the delayed designation of COVID-19-dedicated hospitals and the redistribution of medical staff. A significant amount of labor and resources are required, especially in trauma cases, and as the WHO recognized it as a preventable health issue and an area of focus [2,3], it is a main concern in emergency medical care. In situations where available resources are drastically reduced, such as during the COVID-19 pandemic, understanding changes in the number of trauma patients is extremely crucial.

A study using data from the National Syndromic Surveillance Program (NSSP) discovered that in the first 4 weeks following the declaration of COVID-19 as a national emergency on 13 March 2020, from 29 March to 25 April 2020, there were approximately 42% more emergency department (ED) visits than the previous month [4]. Moreover, a 25% drop in ED visitors from December 2020 to January 2021 was observed compared with the same month a year prior [5]. During the national lockdown in Italy, the number of patients visiting the pediatric ED decreased by 73% to 88% compared with the same period in 2018 and 2019 [6]. Another NSSP-based study found that from mid-March to October 2020, mental health conditions, suicide attempts, all drug and opioid overdoses, intimate partner violence, and child abuse and neglect were higher than in the same period in 2019 [7]. According to Law et al. [8], compared with the previous period, the number of ED visits related to traumatic injuries decreased during the COVID-19 epidemic period. The change in each category was different when the injury pattern was classified as traffic accidents, falls, self-harm and suicide, assaults, and poisonings.

On 20 January 2020, Republic of Korea’s first confirmed case of COVID-19 was documented [9], and the government recommended nationwide social distancing and non-face-to-face activities starting from 22 March 2020 [10]. Republic of Korea has the highest suicide rate among the Organization for Economic Co-operation and Development countries; thus, the use of firearms and narcotics is prohibited [11]. As the detailed patterns of injuries vary greatly by country, this study aimed to identify the patterns of injuries in trauma patients visiting the ED before and after the COVID-19 outbreak. Republic of Korea created the Emergency Department-based Injury In-depth Surveillance (EDIIS) guideline in 2006, and according to this guideline, 23 tertiary emergency medical centers nationwide collect information by tracking all injured patients during the ED visit from hospitalization until discharge. Among the 23 hospitals that collected data used in this study, some hospitals doubled as trauma centers. The collected data was submitted to the Korea Disease Control and Prevention Agency (KDCA), and more than 3 million injury-related data had been collected from 2006 until 2020 [12].

According to the data released by the KDCA, 510 EDs received more than 10 million ED visits nationwide annually from 2016 to 2019, but with less than 8 million ED visits in 2020, which is a significant decrease [13]. For injured patients surveyed through EDIIS in 2020, the number of visits was the lowest since the participation of hospitals was expanded to 23 in 2015 [14]. Statistics showed that during the COVID-19 pandemic, the total number of ED visits and the number of ED visits related to injury decreased, and we aimed to analyze the change in the injury data. During the pandemic, it was critical to redistribute available medical resources immediately and appropriately as additional medical resources were required to treat quarantined patients. This study is expected to provide information that can help medical personnel or hospital beds to be flexibly redistributed.

## 2. Methods

### 2.1. Study Design and Setting

This retrospective cross-sectional study used data from KDCA. The survey evaluated all injured patients who visited the ED of 23 tertiary teaching hospitals throughout the country, and the collected data were investigated according to the EDIIS guidelines. Before the analysis, the data were anonymized [15,16,17].

### 2.2. Data Collection and Variables

The unit of analysis for this study was the number of visits, and a total of 541,515 ED visits were included between 1 January 2018 and 31 December 2020. The study included injured patients over 20 years of age. The years 2018, 2019, and 2020 were referred to as groups 1, 2, and 3, respectively, and 2018 and 2019 were referred to as the “pre-COVID-19” cohort and 2020 as the “COVID-19” cohort.

The EDIIS guidelines include the following investigation items: epidemiologic factors, such as age, sex, insurance type; date of visit; date of injury; location of injury; activity at the time of injury; mechanism of injury; and treatment outcome. This study evaluated the length of ED stay, ED disposition, admission duration, and admission result through processing data.

### 2.3. Statistical Analysis

In this study, the Chi-square test and Fisher’s exact test were employed when the dependent variable was a categorical variable. When the dependent variable was a continuous variable, analysis of variance (ANOVA) was performed comparing groups 1, 2, and 3, while the Student’s *t*-test was used to compare the pre-COVID-19 and the COVID-19 groups. For the post hoc test, Bonferroni’s correction was used in ANOVA. Statistical significance was wet at a two-tailed *p*-value < 0.05. The *p* values for each of the analyses comparing groups 1, 2, and 3 and the pre-COVID-19 and the COVID-19 groups are presented in Table 1. The Statistical Package for the Social Science version 26 was used for the statistical analysis.

## 3. Results

During the study period, a total of 768,273 visits to 23 emergency medical centers due to traumatic injury were documented. Data from 541,515 ED visits were obtained, excluding those under the age of 20 years. This study analyzed the pattern of injuries in 2018 and 2019 before the onset of the COVID-19 pandemic and 2020 during the pandemic.

### 3.1. Demographic Properties

Data on the number of ED visits related to trauma, disaggregated by various characteristics such as year, sex, mode of ED arrival, and insurance type are presented in Table 1. Prior to the COVID-19 pandemic, there were over 190,000 trauma-related ED visits each year, with 196,798 visits in 2018 and 192,996 visits in 2019. However, during the COVID-19 period, the number of visits decreased to 151,721, representing a decline of more than 40,000 visits.

In terms of sex, males accounted for approximately 56% of all ED visits, while females accounted for 44% in all groups. The proportion of ED visits conducted via 911 gradually increased over time, from 30.3% in 2018 to 34.5% in 2020. National health insurance was the most common type of insurance used for ED visits, accounting for 77.5% of visits in 2018, 78.0% in 2019, and 79.9% in 2020. The proportion of ED visits covered by national health insurance gradually increased over time. The second most common type of insurance used for ED visits was automobile insurance, which accounted for 13.4% of visits in 2018, 13.0% in 2019, and 11.0% in 2020.

### 3.2. Alcohol Positivity and Intentionality

Alcohol-related injuries refer to injuries sustained by patients while under the influence of alcohol, or injuries caused by an intoxicated individual. The incidence of alcohol-related injuries has decreased during the COVID-19 pandemic, with 24,806 cases in 2018, 26,298 cases in 2019, and 20,951 cases in 2020 (Table 1). However, the proportion of alcohol-related injuries increased from 12.6% in 2018 and 13.6% in 2019 to 13.8% in 2020, the year of the COVID-19 pandemic. While the total number of traumas decreased by more than 40,000 during the pandemic year, the number of self-harm and suicide injuries remained constant year after year, with 8288 cases in 2018, 8917 cases in 2019, and 8911 cases in 2020. Moreover, the proportion of self-harm and suicide injuries increased from 4.2% in 2018 and 4.6% in 2019 to 5.9% in 2020.

### 3.3. Injury Profile

Falls and slips are the most commonly reported cause of injury, which includes cases of slipping or falling from the ground or stairs, and the incidence rate increased gradually over the survey period (60,798 [30.9%] in 2018; 61,286 [31.8%] in 2019; and 49,156 [32.4%] in 2020; see Table 1). The next most common cause was traffic accidents, which included pedestrian and motor vehicular accidents. The incidence rate was lowest during the period of the COVID-19 pandemic (37,733 [19.2%] in 2018; 36,332 [18.8%] in 2019; and 27,144 [17.9%] in 2020). Collision injuries included damage caused by collisions and bumps with objects, people, and animals, which excluded contact with machinery. The incidence rate for this cause decreased during the period of the COVID-19 pandemic (33,360 [17.0%] in 2018; 33,143 [17.2%] in 2019; and 24,254 [16.0%] in 2020). Penetration, which includes scratches, cuts, tears, piercings, bites, or stings, showed an increased incidence rate during the period of the COVID-19 pandemic (25,375 [12.9%] in 2018; 24,295 [12.6%] in 2019; and 20,269 [13.4%] in 2020). Poisoning included cases of intentional or unintentional intoxication with therapeutic and non-therapeutic drugs or toxic substances, with an increased incidence rate during the period of the COVID-19 pandemic (7834 [4.0%] in 2018; 7859 [4.1%] in 2019; and 7167 [4.7%] in 2020). Substance exposure referred to exposure to chemicals, foreign substances, electricity, radiation, sound, and vibration, and its incidence rate remained constant regardless of the presence of COVID-19 (956 [0.5%] in 2018; 962 [0.5%] in 2019; and 791 [0.5%] in 2020). The incidence rates of drowning, hanging, and asphyxia remained constant across the three groups (910 [0.2%] in 2018; 988 [0.2%] in 2019; and 961 [0.2%] in 2020), whereas the rate of thermal injury gradually decreased (3784 [1.9%] in 2018; 3389 [1.8%] in 2019; and 2452 [1.6%] in 2020). Machine-related injuries, excluding vehicles, showed an increased incidence rate during the period of the COVID-19 pandemic (2403 [1.2%] in 2018; 2363 [1.2%] in 2019; and 2140 [1.4%] in 2020), whereas stress-induced injuries decreased during the period of the COVID-19 pandemic (8488 [4.3%] in 2018; 8456 [4.4%] in 2019; and 5781 [3.8%] in 2020). Figure 1 showed the monthly trend in the incidence of the injury of interest in this study, which was closely examined during the period of the COVID-19 pandemic.

The most common site of injury was the house, and the injury rate increased during the period of the COVID-19 pandemic (67,737 [34.2%] in 2018; 67,311 [34.9%] in 2019; and 58,975 [38.9%] in 2020; see Table 1). The road was the second most common location for injury, and the rate of injury on the road gradually decreased (60,335 [30.7%] in 2001; 58,630 [30.4%] in 2019; and 43,225 [28.5%] in 2020). Commercial facilities are the third most dangerous location, with incidence rates increasing during the COVID-19 pandemic (21,303 [10.8%] in 2018; 22,988 [11.9%] in 2019; and 17,520 [11.5%] in 2020). The rate of injury prevalence in factories and industrial facilities has continued to fall (14,226 [7.2%] in 2018; 12,546 [6.5%] in 2019; and 9208 [6.1%] in 2020). Prior to the COVID-19 pandemic, more injuries occurred outdoors (indoor vs. outdoor; 96,150 [48.9%] vs. 99,199 [50.4%] in 2018; and 95,778 [49.6%] vs. 96,217 [49.9%] in 2019); however, during the period of the pandemic, more injuries occurred indoors (indoor vs. outdoor; 79,594 [52.5%] vs. 71,137 [46.9%] in 2020). In 2018, most injuries occurred in the office (24,380 [12.4%]), but during the pandemic years of 2019 and 2020, most injuries occurred in the room and bedroom (23,678 [12.3%] in 2019; 20,863 [13.8%] in 2020).

During the period of the study, patients were most frequently engaged in activities of daily living at the time of injury (82,432 [41.9%] in 2018; 81,216 [42.1%] in 2019; and 61,030 [40.2%] in 2020). In 2018, the second most common location of injuries occurred at work (3007 [15.2%]), but in 2019 and 2020, injuries during leisure activities were the second most common (31,132 [16.1%] in 2019 and 27,114 [17.9%] in 2020).

### 3.4. Outcomes of Medical Care

For both pre-COVID-19 and during the COVID-19 pandemic, more injuries occurred on weekdays than on weekends (weekday vs. weekend; 100,813 [51.2%] vs. 95,985 [48.8%] in 2018; 97,964 [50.8%] vs. 95,032 [49.2%] in 2019; and 77,226 [50.9%] vs. 74,495 [49.1%] in 2020; see Table 1). The highest number of injuries occurred in the evening between 15:00 and 22:59 (88,863 [45.2%] in 2018; 87,174 [45.2%] in 2019; and 68,166 [44.9%] in 2020).

In 2018, the time from ED admission to discharge was 211.96 ± 527.66 min (126.00 min [interquartile range {IQR} 160]); in 2019, it was 206.63 ± 341.04 min (129.00 min [IQR 164]); and in 2020, it was 213.57 ± 1381.47 min (132.00 min [IQR 173]), with a significant difference observed in the post hoc testing of the length of ED stay between groups 2 and 3. The most common outcome after treatment in the ED was being discharged (including voluntary discharge; 153,923 [78.2%] in 2018; 152,106 [78.8%] in 2019; and 117,102 [77.2%] in 2020), and hospitalizations in the intensive care unit (ICU) or ward were highest during the COVID-19 period (18,938 [12.5%] admitted to ward; 9560 [6.3%] admitted to ICU). Furthermore, the ED mortality rate was 0.7% in 2018, 0.6% in 2019, and 0.8% in 2020. 

The average length of hospital stay had been 15.53 ± 21.69 days (9.49 days [IQR 13.52]) in 2018, 14.21 ± 18.52 days (8.84 days [IQR 12.72]) in 2019, and 14.22 ± 18.52 days (8.75 days [IQR 12.50]) during the COVID-19 pandemic in 2020, with significant differences observed in the post hoc analysis of length of hospital stay between groups 1 and 2, as well as between groups 1 and 3. The mortality rate among hospitalized patients had been 4.1% in 2018, 4.0% in 2019, and 4.2% during the COVID-19 pandemic in 2020.

## 4. Discussion

The COVID-19 pandemic had a significant impact on patient visits to the ED. Patients with COVID-19 were transported to the hospital via the ED, and severe COVID-19 patients required substantial labor, resources, and care time. Medical staff on the frontlines complained about the lack of medical resources, but statistically, the total number of patients visiting the ED decreased significantly, as it did in Republic of Korea [4,5,18,19,20,21]. According to statistical data from 167 emergency medical institutions above the center level provided by the National Emergency Department Information System, the number of patients visiting the ED was 5,998,742 in 2018 (153 institutions, disease group vs. non-disease group; 4,193,813 vs. 1,587,926) and 6,146,688 in 2019 (162 institutions, disease group vs. non-disease group; 4,280,200 vs. 1,644,094). However, in 2020, it decreased to 4,801,361 (167 institutions, disease group vs. non-disease group; 3,325,207 vs. 1,308,621). Many studies have reported that, while the total number of ED users has decreased, the pattern of increase and decrease by patient group differs [7,8,22,23,24,25,26]. Hence, this study aims to examine the changes in the pattern of preventable injury during the COVID-19 period.

During the COVID-19 pandemic, the total number of injured patients visiting the ED decreased, except for January. The lower peak in the total number of injuries, especially in March, September, and December, seemed to mirror the COVID-19 infection peaks in Republic of Korea during those months (Figure 1). This could be attributed to patients’ perception at the time that hospitals were riskier due to the presence of infectious disease patients and voluntary activity restrictions due to concerns regarding COVID-19. Thus, the actual number of injuries decreased and patients who required emergency medical care did not visit the ED. Furthermore, in the monthly trend analysis according to the detailed injury types, it was found that injuries related to motor vehicular accidents, falls and slips, assaults, and poisonings also decreased during the COVID-19 period. However, self-harm and suicide showed a similar monthly trend compared to the pre-COVID-19 period (Figure 2). These findings suggested that the decrease in the total number of injuries during the COVID-19 pandemic was not uniform across all injury types, and it highlights the importance of monitoring the changes in injury patterns during public health emergencies.

In the age- and sex-related injury data analysis, we found that individuals under the age of 10 experienced the highest frequency of injuries, with 22.0%, 19.7%, and 21.9% occurring in 2018, 2019, and 2020, respectively. The injury rate decreased annually for individuals under the age of 20, with 22.0% and 8.7% of injuries occurring in the 0–9 and 10–19 age groups in 2018, 21.8% and 8.6% in 2019, and 19.7% and 7.0% in 2020. Interestingly, we observed an increase in the percentage of injuries among individuals in their 20s and 60s each year, with 12.7% of injuries occurring in the 20–29 age group, 9.2% occurring in the 60–69 age group, and 11.7% occurring in the ≥70 age group in 2018; 12.8%, 9.8%, and 12.3% in 2019; and 13.8%, 10.8%, and 13.4% in 2020, respectively (refer to Appendix A, Table A1 and Figure A1 for more details).

Similar to previous studies [8,27], this study also found a decrease in motor vehicle-related injuries during the COVID-19 pandemic (Table 1). However, an increase in the rate of motor vehicle-related injuries among people in their 20s and 40s was observed in the age-specific analysis (Appendix A, Table A1). Males were more likely to be injured in motor vehicles (64.5% in 2018; 62.2% in 2019; and 66.0% in 2020), and the highest rate was observed among those in their 20s (16.2% in 2018; 16.6% in 2019; and 17.9% in 2020). In the age group of 0–9 years, the annual rate of motor vehicle-related injuries decreased (7.0% in 2018; 6.2% in 2019; and 5.3% in 2020).

A report released by Republic of Korea’s Ministry of Land, Infrastructure and Transport indicated that the volume of traffic in metropolitan areas decreased by 12.1% compared to 2019, with public transportation usage decreasing by 26.8% [28]. The report also found that the decrease in public transportation use had been greater than the decrease in overall wide-area traffic, and this had been due to people choosing to drive instead of using public transportation. According to the report, traffic decreased by 11.6% in the work area but increased by 21% and 18.9% in the tourist and commercial areas, respectively. This was because telecommuting reduced commuting and increased time spent at home during existing leisure time, resulting in a decrease in traffic. As a result, the number of motor vehicle-related accidents decreased during the COVID-19 pandemic.

Although there have been reports of COVID-19 patients experiencing fainting and falling [29,30], the overall number of fall- and slip-related injuries decreased during the pandemic. Prior to the pandemic, the most common accidents among those aged 0–9 years were falls and slips (27.2% in 2018 and 27.1% in 2019); however, during the COVID-19 period, they were most frequent in individuals aged 70 years or older (25.2% in 2020). The decrease in outdoor activity during the pandemic resulted in a decrease in overall fall- and slip-related injuries, as outdoor injury rates had been higher before the pandemic. However, the rate of falls and slips in the elderly (aged 60 years or older) increased, as did the rate of injuries occurring in rooms and bedrooms. This suggests that the decrease in outdoor activity has resulted in a decline in physical activity and deterioration in physical functioning, leading to an increase in the rate of injury in the elderly population who are vulnerable to falls [31].

The results suggest that the proportion of self-harm and suicide in females increased annually and reached the highest point during the COVID-19 pandemic (54.8% in 2018; 57.0% in 2019; and 60.6% in 2020). Self-harm and suicide were most prevalent in the 20–29 age group, especially during the COVID-19 period (22.7% in 2018; 24.6% in 2019; 29.3% in 2020). Research indicates that when a disaster occurs, short-term and long-term mental health problems are not only experienced by victims but also by the general public [32,33,34,35,36]. Although the total number of injured patients decreased during the pandemic, the rate of self-harm and suicide-related injuries increased. Notably, the number of ED visits for self-harm and suicide among patients aged 20–29 increased significantly during the pandemic compared to previous years (age 20–29 years: 2148 in 2018; 2515 in 2019; and 3014 in 2020). Data suggest that self-harm and suicide may have increased during the COVID-19 pandemic, as the incidence of ED visits for these types of injuries remained constant despite an overall decrease in ED visits for other types of injuries.

Several studies have consistently reported that during the COVID-19 pandemic period, a decrease in domestic violence, intimate partner violence, and suspected child abuse and neglect was observed [7,8,24,37]. However, one study reported an increase in occupational sexual assault [38]. This study found that violence decreased during the COVID-19 period, including the number of patients visiting the ED across all age groups. Assault was more common in men (65.8% in 2018; 66.3% in 2019; and 63.5% in 2020), with the highest incidence in individuals aged 20–29, particularly during the COVID-19 period (25.2% in 2018; 24.4% in 2019; and 25.8% in 2020). This study highlighted the intentionality of injuries, and it is unclear whether the decrease in assault was due to reduced exposure to violence or to a circumstance where victims of violence were unable to seek medical attention, even if violence had increased.

During the COVID-19 pandemic, the incidence of poisoning had decreased in all age groups except for those aged 20–29 years when compared to the previous year. However, the proportion of poisoned patients increased during the pandemic period (4.0% in 2018, 4.1% in 2019, and 4.7% in 2020). Considering that poisoning is often associated with self-harm and suicide (5799 patients [62.4%] in 2018, 6202 patients [67.5%] in 2019, and 6029 patients [73.5%] in 2020), it is plausible that the increase in poisoning rates is related to the rise in self-harm and suicide rates. Females were more likely to be poisoned, particularly during the COVID-19 period (55.2% in 2018; 57.1% in 2019; and 60.1% in 2020). In 2018, those aged 40–49 years had the highest poisoning injury rate (15.8%); however, in 2019 and 2020, those aged 20–29 years had the highest poisoning rate (17.8% in 2019 and 21.7% in 2020). Although some studies have reported an increase in opioid overdose during the COVID-19 pandemic [7,22], this study only confirmed four records of opioid overdose out of 26,784 patients whose overdosed substance contents were confirmed (out of 768,273 injured patients of all ages). As the distribution of opioids is illegal in Republic of Korea, there is no separate checkbox for opioids and additional input is required. Failure to actively input the information in the remarks section could result in the omission of opioid exposure data, leading to an underestimation of the number of drug users.

In conclusion, our data suggest that there had been a decrease in the overall number of trauma patients during the COVID-19 pandemic. However, whether this decrease was attributed to fewer actual injuries or if patients simply did not seek medical care due to fear of COVID-19 remains unclear. Notably, the rates of suicide attempts remained similar, indicating a potential increase in self-harm or suicidal behavior during the pandemic. Furthermore, our data showed that patients who did seek medical care during the pandemic tended to have more severe injuries, as evidenced by the increase in hospitalizations and ICU admissions. This highlights the continued need for ED staff to provide high-quality care to trauma patients, while also managing the demands of the pandemic response.

A delicate and adaptable approach is required, given the reduced medical personnel and resources available, to effectively manage ED operations during the pandemic. It is important to prioritize the mental health needs of both patients and healthcare workers, with increased support and resources provided for those at risk for self-harm or suicide attempts. Healthcare workers should also be provided with the necessary tools and resources to manage the increased stress and workload caused by the pandemic.

In summary, the COVID-19 pandemic highlighted the ongoing need for ED services to provide care to patients with medical emergencies unrelated to the virus.

### Limitation

Establishing a correlation between variations in patient visit patterns and social distancing is challenging, as social distancing was implemented and adjusted differently in each region based on the number of COVID-19 patients. Further investigation is required to determine if the reduction in the number of emergency department visits is due to a decline in the actual number of injured patients or fear of hospitals. Additionally, analyzing injuries that are constantly changing due to the COVID-19 spread is challenging as the EDIIS utilized in this study is not available in real-time. 

## 5. Conclusions

As a result, our analysis suggested that the COVID-19 pandemic had a significant impact on the number and severity of trauma patients seeking medical care. While the overall number of trauma cases decreased during the pandemic, the number of patients requiring hospitalization or ICU admission increased, indicating that those who did seek medical care tended to have more severe injuries. This highlighted the continued need for the ED staff to provide high-quality care to trauma patients while balancing the demands of the pandemic response.

It was also noteworthy that the suicide attempt rates remained constant during the pandemic, suggesting that there may have been an increase in self-harm or suicidal behavior among vulnerable individuals. Targeted care and support for these patients should be a priority, as well as increased resources and support for healthcare workers who are facing increased stress and workload.

Amidst the COVID-19 pandemic, EDs had to continue to provide care to patients with medical emergencies unrelated to the virus. Effective management of existing patients while also treating COVID-19 patients or those suspected of being infected requires a delicate and adaptable approach to ED operations due to the reduced medical personnel and resources available.

## Figures and Tables

**Figure 1 healthcare-11-01483-f001:**
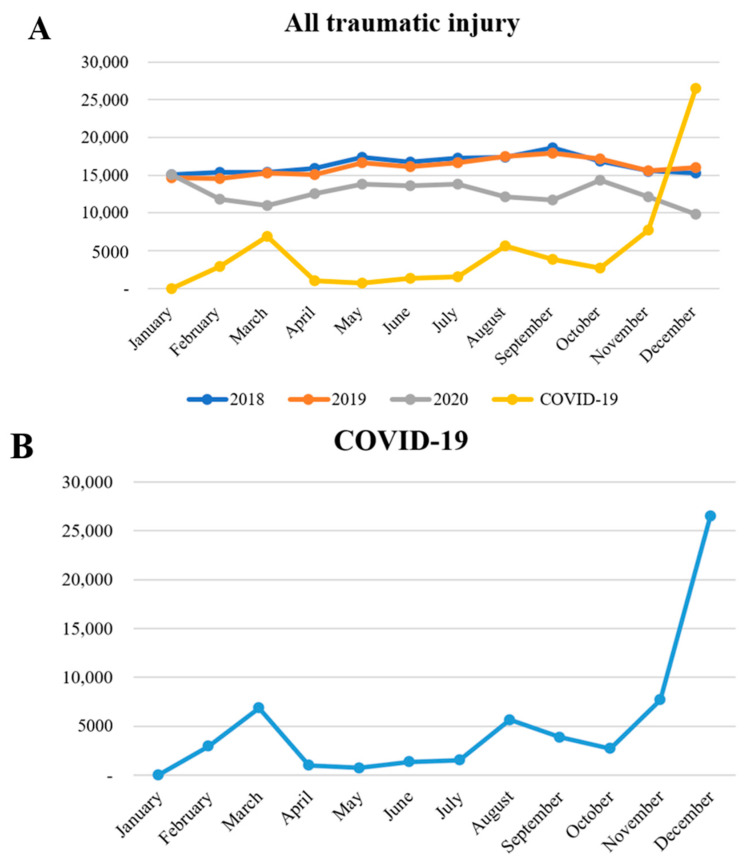
Trends in monthly overall Injuries by year and monthly COVID-19 case counts in 2020. The total number of injured patients, in particular, had shown lower peaks in March, September, and December (**A**). This was attributed to the fact that the waves of COVID-19 infections in Republic of Korea had occurred on 18 February, 18 August, and 31 November (**B**). Social distancing had been officially implemented in Republic of Korea on 22 March and remained in effect until 18 April 2022, implying that the number of patients had decreased even before social distancing was implemented, mirroring the course of the COVID-19 pandemic.

**Figure 2 healthcare-11-01483-f002:**
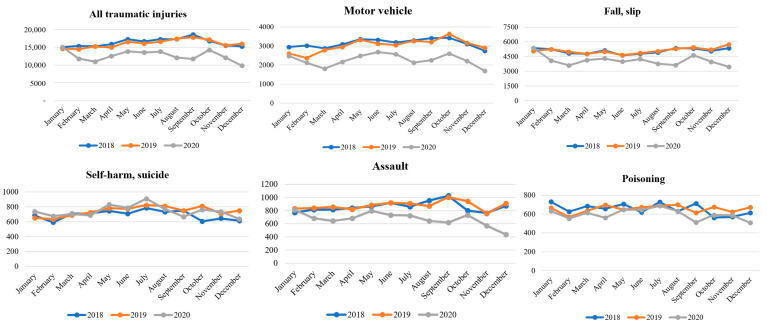
Monthly estimated injury-related emergency department visits by type of injury, Republic of Korea (1 January 2018 to 31 December 2020). When analyzing the trend by month, the total number of traumatic injuries had decreased in 2020, as had the number of injuries related to motor vehicles, falls and slips, assault, and poisoning.

**Table 1 healthcare-11-01483-t001:** Epidemiologic and clinical characteristics of patients with traumatic injury visiting the emergency room.

		Pre-COVID-19				COVID-19		Total		*p* Value ^a^	*p* Value ^b^
		2018 (Group 1)		2019 (Group 2)		2020 (Group 3)					
		*N*	%	*N*	%	*N*	%	*N*	%		
Number of patients		196,798	36.3	192,996	35.6	151,721	28.0	541,515	100		
Age (yr) mean ± *SD*		49.54 ± 18.491 ^c^		50.02 ± 18.748 ^d^		50.3 ± 19.047 ^e^		49.92 ± 18.742		<0.001	<0.001
median (IQR)		49.00 (29)		50.00 (30)		50.00 (30)		50.00 (29)			
Sex										0.026	0.008
	Male	111,132	56.5	108,860	56.4	86,236	56.8	306,228	56.6		
Mode of arrival										<0.001	<0.001
	Walk-in (include car, foot, etc.)	2125	1.1	2708	1.4	3021	2.0	7854	1.5		
	911	59,723	30.3	62,296	32.3	52,376	34.5	174,395	32.2		
	Ambulance	13,613	6.9	12,636	6.5	9757	6.4	36,006	6.6		
	Others	121,337	61.7	115,356	59.8	86,567	57.1	323,260	59.7		
Insurance										<0.001	<0.001
	National health insurance	152,546	77.5	150,490	78.0	121,299	79.9	424,335	78.4		
	Self-pay (uninsured)	9437	4.8	8233	4.3	5867	3.9	23,537	4.3		
	Car insurance	26,319	13.4	24,995	13.0	16,644	11.0	67,958	12.5		
	Medicaid	7861	4.0	8171	4.2	7150	4.7	23,182	4.3		
	Others	635	0.3	1107	0.6	761	0.5	2493	0.5		
Alcohol-related injury										<0.001	<0.001
	Yes	24,806	12.6	26,298	13.6	20,951	13.8	72,058	13.3		
Intentionality										<0.001	<0.001
	Unintentional	177,590	90.2	173,003	89.6	134,212	88.5	484,805	89.5		
	Self-harm, suicide	8288	4.2	8917	4.6	8911	5.9	26,116	4.8		
	Assault	10,299	5.2	10,528	5.5	8067	5.3	28,894	5.3		
	Others	187	0.1	142	0.1	172	0.1	501	0.1		
	Unknown	434	0.2	406	0.2	359	0.2	1199	0.2		
Mechanisms										<0.001	<0.001
	Traffic accident	37,733	19.2	36,332	18.8	27,144	17.9	101,209	18.7		
	Fall, slip	60,798	30.9	61,286	31.8	49,156	32.4	171,240	31.6		
	Collision	33,360	17.0	33,143	17.2	24,254	16.0	90,757	16.8		
	Penetration	25,375	12.9	24,295	12.6	20,269	13.4	69,939	12.9		
	Poisoning	7834	4.0	7859	4.1	7167	4.7	22,860	4.2		
	Substance exposure	956	0.5	962	0.5	791	0.5	2709	0.5		
	Drowning, hanging, and asphyxia	910	0.2	988	0.2	961	0.2	2859	0.5		
	Thermal injury	3784	1.9	3389	1.8	2452	1.6	9625	1.8		
	Machine	2403	1.2	2363	1.2	2140	1.4	6906	1.3		
	Natural disaster	6	0.0	8	0.0	12	0.0	26	0.0		
	Stress-induced	8488	4.3	8456	4.4	5781	3.8	22,725	4.2		
	Others	15,151	7.7	13,915	7.2	11,594	7.6	40,660	7.5		
Place										<0.001	<0.001
	Road	60,335	30.7	58,630	30.4	43,225	28.5	162,190	30.0		
	House	67,737	34.2	67,311	34.9	58,975	38.9	193,623	35.8		
	Residential facilities	2241	1.1	1957	1.0	1503	1.0	5701	1.1		
	Medical facilities	4824	2.5	4569	2.4	3776	2.5	13,169	2.4		
	School, educational facilities	831	0.4	810	0.4	381	0.3	2022	0.4		
	Sport facilities	6536	3.3	6654	3.4	3835	2.5	17,025	3.1		
	Factory, industrial facilities	14,226	7.2	12,546	6.5	9208	6.1	35,980	6.6		
	Transportation areas, except road	3275	1.7	3228	1.7	2154	1.4	8657	1.6		
	Amusement and cultural public facilities	3068	1.6	2824	1.5	1780	1.2	7772	1.4		
	Farm	2816	1.4	2732	1.4	2385	1.6	7933	1.5		
	Outdoor, sea, river	8424	4.3	7696	4.0	5909	3.9	22,029	4.1		
	Commercial facilities	21,303	10.8	22,988	11.9	17,520	11.5	61,811	11.4		
	Others	1582	0.8	951	0.5	1070	0.7	3603	0.7		
Indoor/outdoor										<0.001	<0.001
	Indoor	96,150	48.9	95,778	49.6	79,594	52.5	271,522	50.1		
	Outdoor	99,199	50.4	96,217	49.9	71,137	46.9	266,553	49.2		
	Unknown	1449	0.7	1001	0.5	990	0.7	3440	0.6		
Detailed places within the building and adjacent areas										<0.001	<0.001
	Toilet, bathroom	8000	4.1	8115	4.2	6853	4.5	22,968	4.2		
	Kitchen	12,374	6.3	11,958	6.2	10,588	7.0	34,920	6.4		
	Living room	15,878	8.1	15,204	7.9	12,241	8.1	43,323	8.0		
	Room, bedroom	23,227	11.8	23,678	12.3	20,863	13.8	67,768	12.5		
	Office	24,380	12.4	23,417	12.1	19,011	12.5	66,808	12.3		
	Stairs	7783	4.0	7941	4.1	5988	3.9	21,712	4.0		
	Garden, yard	2292	1.2	2179	1.1	1818	1.2	6289	1.2		
	Sport facilities	2817	1.4	3208	1.7	2122	1.4	8147	1.5		
	Playground, gym	4239	2.2	4073	2.1	2218	1.5	10,530	1.9		
	Others	95,808	48.7	93,223	48.3	70,019	46.1	259,050	47.8		
Activity										<0.001	<0.001
	Leisure activities	29,094	14.8	31,132	16.1	27,114	17.9	87,310	16.1		
	Daily living activities	82,432	41.9	81,216	42.1	61,030	40.2	224,678	41.5		
	Unpaid labor	25,394	12.9	21,041	10.9	15,728	10.4	62,163	11.5		
	Work	3007	15.2	29,901	15.5	23,801	15.7	83,709	15.5		
	Others	29,871	15.2	29,706	15.4	24,048	15.9	83,625	15.4		
Day of presentation										0.012	0.528
	Weekday (Mon–Thu)	100,813	51.2	97,964	50.8	77,226	50.9	276,003	51		
Time of presentation										<0.001	<0.001
	Day (7:00–14:59)	63,979	32.5	61,953	32.1	48,222	31.8	174,154	32.2		
	Evening (15:00–22:59)	88,863	45.2	87,174	45.2	68,166	44.9	244,203	45.1		
	Night (23:00–6:59)	43,956	22.3	43,869	22.7	35,333	23.3	123,158	22.7		
Length of ED stay (min) mean ± *SD*		211.96 ± 527.66 ^c,d^		206.63 ± 341.04 ^c^		213.57 ± 1381.47 ^d^		210.51 ± 823.01		<0.030	0.088
median (IQR)		126.00 (160)		129.00 (164)		132.00 (173)		128.00 (166)			
Disposition of ED										<0.001	<0.001
	Discharge	153,923	78.2	152,106	78.8	117,102	77.2	423,131	78.1		
	Admission to ICU	10,167	5.2	10,288	5.3	9560	6.3	30,015	5.5		
	Admission to ward	23,818	12.1	21,837	11.3	18,938	12.5	64,593	11.9		
	Transfer	7008	3.6	6704	3.5	4394	2.9	18,106	3.3		
	Expire	1286	0.7	1212	0.6	1242	0.8	3740	0.7		
	Others	596	0.3	849	0.4	485	0.3	1930	0.4		
Admission duration (day) mean ± *SD*		15.53 ± 21.17 ^c^		14.21 ± 18.83 ^d^		14.22 ± 18.52 ^d^		14.688 ± 19.93		<0.001	<0.001
median (IQR)		9.49 (13.52)		8.84 (12.72)		8.75 (12.50)		8.960 (12.86)			
Result of admission										<0.001	<0.001
	Discharge	24,676	72.6	22,812	71.0	20,202	70.9	67,690	71.5		
	Transfer	7868	23.2	7790	24.2	7039	24.7	22,697	24.0		
	Expire	1408	4.1	1293	4.0	1194	4.2	3895	4.1		
	Others	33	0.1	230	0.7	63	0.2	326	0.3		

^a^ Comparing groups 1, 2, and 3. ^b^ Comparing pre-COVID-19 and during COVID-19. ^c,d,e^ The same letters indicate non-significant differences between groups based on Bonferroni correction.

## Data Availability

Data are unavailable due to privacy and ethical restrictions.

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
