# Peer review of "The Pattern of Injuries in the Emergency Room during the COVID-19 Pandemic"

_healthcare, 2023, doi:10.3390/healthcare11101483_

Round 1

Reviewer 1 Report

Buen día.

This is an interesting topic, but this article highlights several concerns, especially regarding the theoretical background, some analytical questions, implications, etc. It does not state the methodology used for the results obtained and therefore does not allow the results to be reviewed. Contributions to this research should be noted in the abstract.

Author Response

This is an interesting topic, but this article highlights several concerns, especially regarding the theoretical background, some analytical questions, implications, etc. It does not state the methodology used for the results obtained and therefore does not allow the results to be reviewed.  

  • As you mentioned, the description of the method section was insufficient; hence, the statistical analysis method used in the study was described in more detail:

In this study, the chi-square test and Fisher’s exact test were employed when the dependent variable was a categorical variable. When the dependent variable was a continuous variable, analysis of variance (ANOVA) was performed comparing groups 1,2, and 3, while the Student’s t-test was used to compare the pre-COVID-19 and the COVID-19 groups. For the post-hoc test, the Bonferroni’s correction was used in ANOVA. Statistical significance was set at a two-tailed p-value <0.05. The p-values for each of the analyses comparing groups 1, 2, and 3 and the pre-COVID-19 and COVID-19 groups are presented in Table 1.

Contributions to this research should be noted in the abstract.

  • This paper not only uncovers the changing patterns of patient data during the pandemic, but further highlights the importance of flexible and sensitive emergency room operations during the pandemic. At the end of the abstract, details on the significance of the paper were added:

Abstract: Conclusion: The COVID-19 pandemic led to a decline in the overall number of trauma patients seeking medical care; however, the proportion of patients requiring hospitalization or intensive care unit admission increased, indicating more severe injuries among those who did seek care. Suicide attempt rates remained unchanged, highlighting the need for targeted care and support for vulnerable patients. EDs must continue to provide care to patients with medical emergencies unrelated to COVID-19, which requires a delicate and adaptable approach to ED operations. To manage the increased stress and workload caused by the pandemic, increased resources and support for healthcare workers are also warranted.

Reviewer 2 Report

The manuscript is mostly a descriptive statistical analysis of emergency room data statistics over 541,515 patients seen in 23 hospitals in South Korea. The authors wish to investigate how emergency room injury trends changed before and after COVID-19. The main finding of the paper is that although the total number of injuries decreased during the pandemic, as a result of the lockdown policies adopted in South Korea, the severity of the injuries increased. This is an interesting empirical result and worth publishing.

The "statistical analysis" subsection of section 2 of the paper needs to be improved. The authors should explain in that section how they organized their data in terms of group 1, 2, 3. The authors should also explain that in addition to the descriptive statistics, a p-value was calculated comparing groups 1, 2, 3, and a separate p-value was calculated to compare the pre-covid groups with the post-covid groups. Then the authors should clarify which tests were used for which of the two p-values.

page 3, Line 116: "through 119" seems to be a typo.

Before Section 4 there is some text that seems to be accidentally inserted from the document template, which should be removed.

Overall, the paper is well written, the results are  clearly explained and placed in the context of previous research. The authors do mention the main limitation of the study, which is that it is possible that the total number of injuries didn't actually decrease during the COVID-19 lockdowns, but that instead people avoided visiting the emergency rooms due to fear of infection, and point out that further research is needed to investigate whether this has affected the reported statistics. It is worth noting that although this may affect results showing a decrease in certain categories of injuries, it would not have an effect on the qualitative reports of increase in certain types of injuries, during the covid period. If an increase is observed, in spite of some plausible under-reporting during covid, that increase would still be there, if the under-reporting of injuries is accounted for.

I recommend a minor revision to address the comments above.

Author Response

The manuscript is mostly a descriptive statistical analysis of emergency room data statistics over 541,515 patients seen in 23 hospitals in South Korea. The authors wish to investigate how emergency room injury trends changed before and after COVID-19. The main finding of the paper is that although the total number of injuries decreased during the pandemic, as a result of the lockdown policies adopted in South Korea, the severity of the injuries increased. This is an interesting empirical result and worth publishing.

The "statistical analysis" subsection of section 2 of the paper needs to be improved. The authors should explain in that section how they organized their data in terms of group 1, 2, 3. The authors should also explain that in addition to the descriptive statistics, a p-value was calculated comparing groups 1, 2, 3, and a separate p-value was calculated to compare the pre-covid groups with the post-covid groups. Then the authors should clarify which tests were used for which of the two p-values.

  • As advised, additional sentences were added due to insufficient explanation as follows.

Methods

In this study, the chi-square test and Fisher’s exact test were employed when the dependent variable was a categorical variable. When the dependent variable was a continuous variable, ANOVA was performed comparing groups 1,2, and 3, while the Student’s t-test was used to compare the pre-COVID-19 and the COVID-19 groups. For the post hoc test, the Dunnett T3 method was used in ANOVA. Statistical significance was set at a two-tailed p-value <0.05. The p-values for each of the analyses comparing groups 1, 2, and 3 and the pre-COVID-19 and COVID-19 groups are presented in Table 1.

page 3, Line 116: "through 119" seems to be a typo.

  • Thank you for your comments. As recommended, “119” has been corrected to “”

Before Section 4 there is some text that seems to be accidentally inserted from the document template, which should be removed.

  • Incorrectly inserted texts in the manuscript were deemed an editorial error and were removed accordingly.

Overall, the paper is well written, the results are clearly explained and placed in the context of previous research. The authors do mention the main limitation of the study, which is that it is possible that the total number of injuries didn't actually decrease during the COVID-19 lockdowns, but that instead people avoided visiting the emergency rooms due to fear of infection, and point out that further research is needed to investigate whether this has affected the reported statistics. It is worth noting that although this may affect results showing a decrease in certain categories of injuries, it would not have an effect on the qualitative reports of increase in certain types of injuries, during the covid period. If an increase is observed, in spite of some plausible under-reporting during covid, that increase would still be there, if the under-reporting of injuries is accounted for.

I recommend a minor revision to address the comments above.

  • Our research aimed to use data on the decline in emergency room visits by injured patients nationwide during the pandemic as a basis for preparation for future pandemics, such as redistributing medical resources. As you noted, it is possible to anticipate a further increase in patients who continue to inflict self-injury or to others during this period, or even experience an exacerbation of their condition.

Reviewer 3 Report

Thank you for the opportunity of reviewing this work which I believe aims to investigate an important indirect health impact to health services during the pandemics. 

In my opinion the main issue is that the Methods section is not sufficiently described and major questions arise. Furthermore, the pandemics did not follow the year calendar and it did not have the same impact on health services through the year of 2020 so, the choice of using yearly periods instead of a shorter time frame (i.e. weekly of monthly) and using the total number of injuries instead of the rate per period seems inadequate.

Follow some more comments and suggestions: 

- The study would benefit from english language revision 

- I believe the word Injury should be in plural in the Title 

- A brief description of the EDBIIS would be useful for an international audience 

- the first sentence in the methods section would fit best in the Introduction section

- Page 2 Line 88, I do not understand the sentence: "The EDBIIS [18-20] was used to collected data"

- The last paragraph of the Results section would fit best in the Discussion

- The Discussion is mostly an extension of the Results section lacking a more in depth discussion on the findings and comparing it to previous studies

Author Response

Thank you for the opportunity of reviewing this work which I believe aims to investigate an important indirect health impact to health services during the pandemics. 

In my opinion the main issue is that the Methods section is not sufficiently described and major questions arise. Furthermore, the pandemics did not follow the year calendar and it did not have the same impact on health services through the year of 2020 so, the choice of using yearly periods instead of a shorter time frame (i.e. weekly or monthly) and using the total number of injuries instead of the rate per period seems inadequate.

  • The Methods section had an insufficient explanation; hence, the statistical methods used in the analysis are described in detail:

Method

In this study, the chi-square test and Fisher’s exact test were employed when the dependent variable was a categorical variable. When the dependent variable was a continuous variable, analysis of variance (ANOVA) was performed comparing groups 1,2, and 3, while the Student’s t-test was used to compare the pre-COVID-19 and the COVID-19 groups. For the post-hoc test, the Bonferroni’s correction was used in ANOVA. Statistical significance was set at a two-tailed p-value <0.05. The p-values for each of the analyses comparing groups 1, 2, and 3 and the pre-COVID-19 and COVID-19 groups are presented in Table 1.

  • Furthermore, as the purpose of this study was to examine the trend changes in detailed injury patterns before and after the COVID-19 outbreak, we deemed it feasible to set the time frame at 1-year intervals. Graphs showing the monthly injury trends are attached in Figure 1 and Appendix Figure A1.

Follow some more comments and suggestions: 

- The study would benefit from English language revision 

--> As you suggested, I will proofread the revised text in English.

- I believe the word Injury should be in plural in the Title 

--> I believe your opinion is very reasonable. I have edited the title accordingly.

- A brief description of the EDBIIS would be useful for an international audience 

--> As you mentioned, the text lacked explanation regarding EDBIIS; thus, more details were added regarding EDBIIS to cater to an international audience:

Introduction

(…)
South Korea created the Emergency Department Based Injury In-depth Surveillance (EDBIIS) guideline in 2006, and according to this guideline, 23 tertiary emergency medical centers nationwide will collect information by tracking all injured patients during the ED visit from hospitalization until discharge. The collated data were submitted to the Korea Disease Control and Prevention Agency (KDCA), and more than 3 million injury-related data have been collected from 2006 until present.

- the first sentence in the methods section would fit best in the Introduction section

 --> Following your advice, I moved the first sentence from the Methods section to the Introduction section to clarify the context and improve flow.

- Page 2 Line 88, I do not understand the sentence: "The EDBIIS [18-20] was used to collected data"

--> The meaning of the sentences written in the draft was unclear; hence, this was corrected according to your suggestion:

Method

Study design and setting

This retrospective cross-sectional study used data from KDCA. The survey evaluated all injured patients who visited the ED of 23 tertiary teaching hospitals throughout the country, and the collected data were investigated according to the EDBIIS guidelines. Before the analysis, data were anonymized.

Data collection and variables

(….)

The EDBIIS guidelines include the following investigation items: epidemiologic factors, such as age, sex, insurance type; date of visit; date of injury; location of the injury; activity at the time of injury; mechanism of injury; and treatment outcome. This study evaluated the length of ED stay, ED disposition, admission duration, and admission result through processing data.

- The last paragraph of the Results section would fit the best in the Discussion

--> As per the reviewer’s recommendation, the last paragraph of the Result’s section should have been discussed in the Discussion section. Thus, this has been described in detail in the Discussion section.

- The Discussion is mostly an extension of the Results section lacking a more in depth discussion on the findings and comparing it to previous studies

--> As you pointed out, the content described in the Discussion section of the draft was only an extension of the Results section. Thus, I reviewed it more closely and described it in detail. Because of this comment, I was able to revise the manuscript with more novelty. Thank you.

Round 2

Reviewer 1 Report

Good morning,

The document meets the requested corrections.

Author Response

Thank you once again for taking the time to review our paper.